# Molecular Dynamics Simulations of a Catalytic Multivalent Peptide–Nanoparticle Complex

**DOI:** 10.3390/ijms22073624

**Published:** 2021-03-31

**Authors:** Sutapa Dutta, Stefano Corni, Giorgia Brancolini

**Affiliations:** 1Dipartimento di Scienze Chimiche, Università di Padova, 35131 Padova, Italy; sutapa.dutta@unipd.it; 2Istituto Nanoscienze, CNR-NANO S3, via G. Campi 213/A, 41125 Modena, Italy

**Keywords:** molecular dynamics, multiscale modeling, nanozymes, functionalized metal nanoparticles, peptide

## Abstract

Molecular modeling of a supramolecular catalytic system is conducted resulting from the assembling between a small peptide and the surface of cationic self-assembled monolayers on gold nanoparticles, through a multiscale iterative approach including atomistic force field development, flexible docking with Brownian Dynamics and µs-long Molecular Dynamics simulations. Self-assembly is a prerequisite for the catalysis, since the catalytic peptides do not display any activity in the absence of the gold nanocluster. Atomistic simulations reveal details of the association dynamics as regulated by defined conformational changes of the peptide due to peptide length and sequence. Our results show the importance of a rational design of the peptide to enhance the catalytic activity of peptide–nanoparticle conjugates and present a viable computational approach toward the design of enzyme mimics having a complex structure–function relationship, for technological and nanomedical applications.

## 1. Introduction

Nanozymes [1,2,3,4,5,6,7,8], the artificially engineered functional nanomaterials, exhibit natural enzyme like intriguing catalytic activity and thus have huge applications in disease diagnosis [4,5,6,9], nanomedicine [4,5,6,10], environmental treatment [4], antimicrobial agent [11]. Although the existing natural enzymes demonstrate a potential role in medicine and industry, their functions are often sensitive to high temperature and extreme pH; moreover, high-cost preparation, difficulty in recycling, substrate sensitivity, low operational stability are some obstacles [1,3,4,5,6,7] that hinder their performance. Thus, the era of synthetic enzymes has been encompassing a broad range of scientific research studies; in particular, using polymers, dendrimers, fullerenes, micelles, and nanomaterials, to name a few [4,5,7]. Among them, gold nanoclusters [12,13,14,15] (diameter < 10 nm) are emphasized as an impressive strategy not only because of the inherent optic, electronic, magnetic properties, but also for their easy surface modification, biocompatibility, inertness, low cost; leading to enhanced durability, stability and catalytic activity of nanozymes. In particular [12,13,14,15], gold monolayer protected clusters (Au-MPC), the nanoclusters passivated by self-assembled monolayer of functional thiol mimicking enzymatic complexity along with the functional environment, have manifested exceptional catalytic activity due to multi-valency and cooperative binding.

In this paradigm, usage of short peptide sequence [16,17,18,19,20,21] to adhere on self-assembled thiolated-ligands over the surface of nanocluster or to form self-assembly of cysteine containing peptides over gold cluster as a scaffold to develop minimalist biocatalyst is one of the recently developed design techniques both from theory and experimental perspectives [16,17,18,19,20,21]. As to theoretical studies, important efforts have been directed towards the development of novel Force Field (FF) parameters to correctly describe peptide-capped gold nanoparticles (NP) by means of atomistic simulations. From simulations, observed experimental evidences have been interpreted as the result of changes in peptide conformation due to the gold nanocluster [22], or to the presence of water molecules and structured sulphur interfacial atoms within the peptide monolayer [23], or the protonation/deprotonation state of peptide-capped gold functional groups [24]. It also has been shown that the self-assembly of bio-functionalized gold NPs [25] can be affected by the flexibility, stability and conjugation dynamics of the peptide functional groups which, in turn, is a function of the peptide length. The role of curvature and ligand properties on the hydrophobicity of small gold NPs as well as larger gold NPs (e.g., modeled as planar gold SAMs), has been systematically investigated by atomistic simulations [26,27,28]. Similarly, the role of polymer chain lengths in polymer-capped silver nanoparticles [29,30] has been addressed by means of fully atomistic Molecular Dynamics (MD) simulations. In addition, the role of protonation of side chain arginine moieties on the adsorption of poly-arginine to silver NP as a function of pH has been investigated computationally [31]. Details of the interaction between amyloid β fibrils gold nanoparticles functionalized with ligands with different terminal groups [32,33,34] have been disclosed through MD simulation. Multiscale MD simulations [35,36] have been applied to unravel protein conformation and driving forces in protein–corona complexes formation. Similarly, coarse-grained MD simulations [37,38,39] of plasma proteins interacting with gold nanoparticles have revealed mutual competition among binding affinities, protein concentration dependent adsorption, effects of orientation on induced conformational changes of proteins upon adsorption on NPs. In a recent experimental work [40], the role of peptide topology on the catalytic activity of peptide-NP conjugates has been studied and an approach toward the design of enzyme mimics has been proposed. However, despite many works done in the field, the rational design of enzyme mimics with enhanced catalytic activity still remains largely elusive.

In this work, we perform fully atomistic MD simulations of two peptides of different length, H_1_ (seq: HWDDD) and H_3_ (seq: HHHWDDD) interacting with Au-MPC, namely Au_144_(L)_60_(L = S(CH_2_)_8_NH_2_^+^). The supramolecular system was prepared and characterized experimentally [41] through the self-assembly of the small peptides on the surface of Au-MPCs. Self-assembly is required for the catalysis, since the peptides do not display any catalytic activity in the absence of Au-MPCs. The peptide was chosen to contain three aspartic acid residues (D) at the C-terminal tail to bind to the positively charged surface of Au-MPC (1.8 ± 0.3 nm), one Tryptophan (W) to be used as a fluorescent probe and a different number of Histidine (H_1_ or H_3_ affecting the peptide length) at the N-terminal tail, which are responsible for the catalytic activity. Experimentally, a linear correlation was noticed between the increase in the catalytic rate constant and the number of Histidine present in the sequence (H_1–3_) [41], suggesting the importance of peptide sequence length in modulating catalytic activity of this kind of multivalent system. We investigated this system using atomistic simulations. Molecular simulations can provide an atomistic insight into the interactions between peptides and the surface of Au-MPC, revealing details that cannot be accessed easily through experiments. In particular, quantities such as the spatio-temporal aspects of interactions involved between peptide-Au-MPC, the conformational changes of the peptide induced by the environment and the diffusion dynamics of the peptide over the surface of Au-MPCs can be accessed by atomistic simulations.

The next section reports the details of the used methods. A section of results illustration follows. Conclusions and perspective are finally illustrated.

## 2. Results

### 2.1. MD of Free Peptide and Free Au-MPC in Water

#### 2.1.1. MD of Peptide in Water

RMSD is computed with respect to peptide backbone atoms as well with respect to $C_{\alpha }\;$ atom of the entire peptide, as a function of simulation time; results are shown in Appendix A.

Cluster analysis is performed on trajectories of peptide H_1_ generated over the last 500 ns data: the most populated cluster accounts for 19,456 structures (RMSD cut-off 0.1 nm) and it is showing an elongated conformation of H_1_ (Figure 1a), this conformation being very similar to that obtained after homology modeling (Appendix A). The other two consecutive clusters are present (Figure 1a), having 13,310 and 5261 structures each, in which the H_1_ peptide adopts a “horse shoe shaped” and a new stretched conformation (with a different orientation of terminal residues), respectively. The transition between the first two most probable conformations of this peptide occurs ~10 ps scale, whereas the second one sustains for a smaller time interval ~10–20 ns span continuously in different time windows over the entire simulation period, the first conformation persists consistently for ~100 ns time scale in various time windows. Thus, the peptide in horse-shoe-shaped conformation is not stable over a larger time interval. In the case of H_3_, the two most popular and comparatively different conformations are found, both with the peptide in a stretched conformation with slightly different bending at C terminal as shown by RMSD 2.2 Å (~10,000 and ~7000 number of structures respectively), as shown in Figure 2a.

#### 2.1.2. MD of Au-MPC in Water

Similarly, RMSD for functionalized ligands for Au_144_(L)_60_ is being depicted in Appendix A. The shape and integrity of the nanocluster, conformational fluctuations of the functionalized ligands do not reflect noticeable changes after the simulation with respect to initial configuration. The same clustering protocol (cut-off 0.2 nm) is applied to the Au-MPC, from which a single structure is extracted (Figure 1c) with ~9000 structures over the last 500 ns equilibrated data.

### 2.2. Binding Poses and Binding Energies of Peptide-Au-MPC Complexes

In Table 1, the docking results and the corresponding driving forces for the H_1_ peptide-Au-MPC association complexes are reported for each conformation of the peptide. Unbiased sampling of three different conformations is done in each Brownian dynamics simulation by choosing all of the three conformations randomly during the trajectory [42]. After the solutes’ positions and orientations are updated during the BD propagation step, BD moves to alternative coordinate sets using the adjacent conformations in the list and a Metropolis algorithm is used to accept (or refuse) trial moves. We find that in H_1_, first conformation is visited for 28.4% times of entire sampling, whereas the second one for 46.5%, and that for the third is 25.1%. However, the final results of docking are obtained after clustering analysis performed over BD trajectories for each of three different conformations separately, to identify different binding pose on the Au-MPC, peptide-nanocluster orientations, free energy of binding, relative population in each cluster and corresponding RMSD. One can identify the most popular and energetically favourable cluster corresponding to each single conformation. Similarly, among three different clusters corresponding to three different conformations, the most suitable one can be chosen primarily based on interaction energy.

In all the resulting complexes, in Figure 1b–d, the binding is dominated by electrostatic interaction, whereas the contribution from non-polar (hydrophobic) desolvation energy ($U_{ds}^{h}$) is smaller. However, for peptide conformations 2 and 3, a slight decrease in the electrostatic term is compensated by an increase of the hydrophobic energy term. Among the association complexes, the most populate and energetically more stable complex A1 with peptide conformation 1 (−52.9 kT) (Figure 3a) is selected as a starting point for MD refinement. For this cluster, the contacting residues, namely the peptide residues which lie at distances <3.5 Å from the Au-MPC atoms, are residue 2W, 3D and 4D.

In Table 2, the docking results for H_3_ peptide are reported. We observe that in analogy with H_1_ also for H_3_, electrostatic interactions energy plays a dominant role with respect to electrostatic desolvation and hydrophobic desolvation energy.

The different binding poses are reported in Figure 2b,c and they have comparable binding energies. However, complex D1 obtained with peptide conformation 1 (Figure 3b) not only has the most favorable binding but it is also the most populated cluster among the others, with a binding free energy −55.1 kT. Thus, this complex is further assessed by MD refinement. The peptide residues involved at the contacting distances are 3H, 5D, 7D. Overall, the docking result indicates that H_3_ is able to establish a stronger binding with Au-MPC with respect to H_1_.

### 2.3. MD Simulations of Peptide-Au-MPC Complexes

RMSD for peptides as well as ligands in H_1_ bound Au_144_(L)_60_ system is depicted in Appendix A. RMSD plots for complex (H_3_ bound Au_144_(L)_60_) generated over the simulation are shown in Appendix A.

In the following part, we focus on observables that characterize the relative positions between the peptides and the Au-MPC as well as the changes in their conformations when moving from the free forms in water to the associated forms in the peptide-Au-MPC complex. The relative peptide-cluster distance in each trajectory is tracked using the distance between the COMs of the peptide and the COM of Au-MPC. At first, the coordinates of the COM of the peptide and of the Au core ($r_{COM}^{H1}$ and $r_{COM}^{Au}$ respectively) are computed, then the distance between them $\left(d_{r_{COM}^{H1}-r_{COM}^{Au}}\left(t\right)\right)$ is plotted over the last 1.0 µs of the total 1.5 µs simulation time. For H_1_ peptide, results are reported in Figure 3c and data indicate that the peptide H_1_ approaches the gold cluster in 3 different stages, namely, between 500 and 700 ns the peptide lies quite far (~25 Å) from the Au core, in the time range 700–1000 ns it eventually approaches at closer distances ~17 Å, experiencing a minimum distance of ~15 Å around 1000 ns and in the last stage going from 1000 to 1500 ns it finally gets relaxed within the surface of the monolayer at a distance of ~16 Å. These stages remind of the binding process for peptides on extended surfaces [43,44]. The analysis of these three different dynamic regimes is even more prominent when plotted in polar coordinates (Figure 3d), where instead of $\left(d_{r_{COM}^{H1}-r_{COM}^{Au}}\left(t\right)\right)\;$ we plot the angular coordinate variations $\left(\theta_{r_{COM}^{H1}-r_{COM}^{Au}}\left(t\right)\right)$, $\left(\phi_{r_{COM}^{H1}-r_{COM}^{Au}}\left(t\right)\right).\;$ For the H_3_ peptide, the variation between center of mass of peptide and gold core $\left(d_{r_{COM}^{H3}-r_{COM}^{Au}}\left(t\right)\right)\;$ is reported in Figure 3e and its polar representations $\left(\theta_{r_{COM}^{H1}-r_{COM}^{Au}}\left(t\right)\right)$, $\left(\phi_{r_{COM}^{H1}-r_{COM}^{Au}}\left(t\right)\right)\;$ in Figure 3f as a function of simulation time. Data indicate that unlike the shorter peptide H_1_, H_3_ experience a fast adsorption dynamics towards the Au-MPC monolayer, relaxing at a distance ~16 Å within the first 500 ns and remains settled at the same distance for the last 1.0 µs of the simulation. The data suggest that a slightly different peptide length may affect the initial phases of the peptide dynamics association towards the Au-MPC.

In order to address the peptide conformational changes induced by the binding to the Au-MPC, clustering algorithm was applied to the last 500 ns of the trajectory. As a result of the interaction with the Au-MPC, H_1_ loses its fully stretched conformation observed in the water solvent (see Section 4.3), and it rather prefers to settle in a bit-curved conformation at one end (inset Figure 3c). For peptide H_3_, the structure reported in the inset of Figure 3e shows an important change in the conformation of the peptide upon binding to Au-MPC. In fact, it completely loses its stretch conformation to display a more folded one which mimics a “horse shoe” shape (the figure is the result of a clustering analysis including ~10,000 snapshots generated over equilibrated trajectories, Appendix A).

We further generate SASA of the peptide ($SASA_{H1}^{f})$ in the absence as well as in the presence of the Au-MPC ($SASA_{H1}^{nc})$ along the abovementioned time domains of the association dynamics and we generate the corresponding distributions, as represented by H($SASA_{H1}^{f})$ and H($SASA_{H1}^{nc})$ respectively. Appendix A signifies that H($SASA_{H1}^{f})$ encompasses a broad range of SASA which is similar to nature of H($SASA_{H1}^{nc})$ over 500–700 ns, where the peptide explores the path randomly, finally between 700–1000 ns and 1000–1500 ns H($SASA_{H1}^{nc})$ becomes narrower. Although, quantitatively the SASA values do not differ significantly, they show that the SASA of the peptide gets reduced upon binding. In analogy with the shorter peptide, SASA of H_3_ (Appendix A) in presence of the nanocluster ($SASA_{H3}^{nc})$ gets reduced with respect to free peptide in water $\left(SASA_{H3}^{f}\right)$.

Similarly, we monitor the difference between the radius of gyration of the peptide in water $\left(R_{g}{}_{H1}^{f}\right)$ with its distribution H$\left(R_{g}{}_{H1}^{f}\right)$, and in the presence of the gold cluster $\left(R_{g}{}_{H1}^{nc}\right)$, H$\left(R_{g}{}_{H1}^{nc}\right)$. Appendix A suggests that radius of gyration of the peptide does not get affected by the binding. Also in the case of peptide H_3_ the radius of gyration (Appendix A) H$\left(R_{g}{}_{H3}^{f}\right)$ and H$\left(R_{g}{}_{H3}^{nc}\right)$ remain comparable.

We further look into –to-end distance fluctuation of the peptide; we consider distance between N-Terminal tail (i.e., terminal N atom of the first residue in the sequence) and C-Terminal tail (i.e., terminal C atom of the last residue in the sequence) of the peptide and then we monitor the distance variation in two different environments ($d_{end2end}{}_{H1}^{f})\;$ and ($d_{end2end}{}_{H1}^{nc})\;$ followed by the corresponding distribution H($d_{end2end}{}_{H1}^{f})\;$ and H($d_{end2end}{}_{H1}^{nc}$). This plots gives us quite interesting results. Mean value of $(d_{end2end}{}_{H1}^{f})\;$ ~15 Å with broad distribution justifies the stretched conformation of peptide in its free form as reported in Figure 4a, while in presence of the nanocluster during the first 500–700 ns ($d_{end2end}{}_{H1}^{nc})\;$ undergoes lots of conformational changes with a mean value of ($d_{end2end}{}_{H1}^{nc})\;$ ~12 Å. The peptide over 700–1000 ns resembles $(d_{end2end}{}_{H1}^{f})\;$ ~15 Å and then finally achieves bended conformation after stable binding over 1000–15,000 ns with a mean value of ($d_{end2end}{}_{H1}^{nc})\;$ ~10 Å.

Next, we focus on dihedral fluctuations of the peptide and follow the same protocol to obtain dihedral distribution over simulated trajectories to distinguish between two circumstances. We find that in presence of the nanocluster, peptide loses its flexibility as indicated by some of the representative cases in Figure 4b. The bimodal distribution of dihedral *ψ* of residue 2W, $H_{H1}^{f}\left(\psi \right)$ and $H_{H1}^{nc}\left(\psi \right)\;$demonstrates that *ψ* is more flexible in the free state and up to 1000 ns in presence of the nanocluster, i.e., in the initial period of exploration. Once the peptide forms stable binding on the surface of monolayer, dihedral fluctuates less and $H_{H1}^{nc}\left(\psi \right)$ becomes unimodal. We compute MSD (Mean Squared Displacement) of the center of mass for the peptide as a function of simulation time, $\langle {\left(r_{COM-H1}^{f}\right)}^{2}\left(t\right)\rangle$ (Figure 5a), and we observe that it has a linear dependence over *t* which is characteristic of normal diffusion in liquid and the diffusion coefficient ($D_{H1}^{f}$) is ~0.28 $\times \;10^{-5}$
${\mathrm{cm}}^{2}/s$ (in good agreement with the HYDROPRO value); however, in proximity of constraint gold cluster the peptide shows deviation from normal diffusion as shown by $\langle {\left(r_{COM-H1}^{nc}\right)}^{2}\left(t\right)\rangle \;$in Figure 5b–d. As a result, the magnitude of diffusion coefficient ($D_{H1}^{nc}$) gets reduced eventually over 500–700 ns ~0.08 $\times \;10^{-5}$
${\mathrm{cm}}^{2}/s$, 700–1000 ns and then this is more prominent over 700–1000 ns, followed by 1000–1500 ns. This is a signature of diffusion [45,46] and resembles [47] binding dynamics in bio-molecules.

The induced conformational change of H_3_ peptide is highlighted in Figure 4c. In its free state, the peptide prefers to remain in a fully stretched conformation with a mean value of ($d_{end2end}{}_{H3}^{f})$ ~17 Å, while in its bound form it assumes a more curved conformation along with average value ($d_{end2end}{}_{H3}^{nc})$ ~7 Å. Similarly, the peptide dihedrals lose their flexibility in the bound state (Figure 4d); bimodal dihedral distribution of residue 4 W ($H_{H3}^{f}\left(\psi \right)$) and that of residue 5D ($H_{H3}^{f}\left(\phi \right)$) transfer into unimodal form ($H_{H3}^{nc}\left(\psi \right)$), ($H_{H3}^{nc}\left(\phi \right)$). Besides, MSD plots $\langle {\left(r_{COM-H3}^{f}\right)}^{2}\left(t\right)\rangle$ and $\langle {\left(r_{COM-H3}^{nc}\right)}^{2}\left(t\right)\rangle$ in Figure 5e,f demonstrate that peptide follows normal diffusive motion with diffusion coefficient ($D_{H3}^{f}$) is ~0.19 $\times \;10^{-5}$
${\mathrm{cm}}^{2}/s$ (again in line with HYDROPRO results), while in the vicinity of Au-MPCs it carries non-linear dependence of MSD over time *t*.

Ultimately, we identify the ligands of the Au-MPC forming direct contacts with the peptides. First, we compute distance between all atoms of the peptide and the terminal amine N atom of the ligands and then sort the ones that come ≤3 Å over simulated trajectories. Then, we generate probability of finding the ligands as per contact as $P\left(close_{H1}^{lig}\right)$ (Figure 6a) and then also point out the ligands that come close to the peptide simultaneously over simulated data (Figure 6b and Appendix A). We observe that there are groups of ligands that are initiating the binding over the 500–700 ns time domain forming a transient binding pocket for the peptide, followed by another group of ligands contacting the peptide over the 700–1000 ns stage and finally a group of ligands which belong to a stable binding patch over the last stage. This means that peptide explores a large share of the surface of the monolayer before finding suitable binding location. We also identify that the peptide is forming contact primarily via the carbonyl O atom of 1H, the side chain carboxyl oxygens of 4D and 5D and the terminal carboxyl oxygen of the 6D residues. However, there are some ligands (id: 12, 26, 33, 34, 35, 40) which form strong binding over the entire last 500 ns of the trajectory, possible by forming contacts with multiple residues at the same time (representative cases are in Appendix A) while some of the ligands (3, 21, 56) (Appendix A) form transient weaker binding patch by showing back and forth motion towards peptide spanning a broad distance between 3 to 20 Å over simulation. Next, we probe inter-ligand fluctuations of the ligands belonging to the binding patch for the peptide. We address distance fluctuation between terminal N atoms of those ligands over simulation (Appendix A); they come in proximity with each other with a mean value of distance of ~5 Å. The final binding pose is depicted in Figure 5c. This event resembles studies [48] on formation of transient protein like binding pocket; mimicking protein–ligand recognition mechanism formed by functionalized monolayer protected gold nanocluster or that for designing nanoreceptors with targeted affinity [49].

The same plots are produced for H_3_ to disclose the ligands forming direct contact with H_3_, namely $P\left(close_{H3}^{lig}\right)$ in Figure 6d and to distinguish the ones forming contact points simultaneously (Figure 6e and Appendix A) over the simulation. Here, also 2H, 5D, 6D and 7D residue networks participate in binding across monolayer, followed by a group of ligands manifesting strong interaction (Appendix A) with multiple and consistent contact patches. Some exhibit weaker connectivity (Appendix A). The final binding pose is depicted in Figure 6f. The connectivity between ligands belonging to the binding networks is shown by distance fluctuation over simulation in Appendix A.

In our next analysis, we probe pair correlation function between the S atom of the thiol group and the central Aspartic acid residue of both the peptide sequences ($g_{S-ASP}^{H1-nc}\left(r\right),\;\;g_{S-ASP}^{H3-nc}\left(r\right)$) over simulation data. This gives us a better idea regarding spatial arrangement of peptides around monolayer. We observe that during 700–1000 ns, H_1_ penetrates into a channel formed by adjacent monolayers; however, the peptide does not remain inside the channel for too long; in its stable binding mode it settles over the surface of layer, which is followed by H_3_ in its bound form (Appendix A, snapshots in Appendix A).

The secondary structures content for the free peptide in solution and for the peptide complexed with the nanocluster has been computed using the DSSP in-built algorithm of (VMD) [50]. Appendix A indicates that the overall effect of the Au-MPC on both peptides is to increase the β-turn propensity with respect to the random coil. This effect is more prominent in the case of H_3_. We then compute the stability [51] of the peptides in terms of the number of contacts in the final conformations and in different environments using Go^−^ contact map analysis [52,53]. The Go^−^ coarse grained model is based on a lower resolution description of the peptide in which beads are located at C_α_ position. For the present application, a fully atomistic force field represents a very good balance between accuracy and computational feasibility. For longer peptides, simplified schemes such as coarse grained descriptions (e.g., the recent Go-MARTINI FF for peptide) should also be considered, although such FFs should be extended to the nanocluster as well. More specifically, the analysis is based on contact generated from native structure and interacting through LJ potential and it represents an established method to understand conformational transition in proteins. When the method is applied to our peptides, we find the presence of native contacts, only between residues 1H and 6D of H_3_ peptide in the presence of nanocluster, based on the standard distance threshold and sequence distance cut-off. No contact map is observed in the corresponding free state of the same peptide. This implies the presence of significant changes in the peptide conformation occurring only in the longer peptide sequence and its bound form.

## 3. Discussion

We find that if the peptide has more catalytically active residues in its sequence, it forms stronger, stable and faster binding network across the monolayer. Besides, it also undergoes drastic changes in conformation showing two different but specific orientations for the two peptides in a regime of non-rigid binding to the Au-MPC. H_3_ prefers to remain in horse-shoe-shaped conformation unlike the stretched one and the loss of flexibility is also more prominent here.

Thus, we realize that the surface of the monolayer is the suitable location for formation of stable binding, if the peptide penetrates into cavity formed by functionalized ligands then Histidine will not be favorable for catalysis; the larger the sequence, the lesser the probability to get buried inside, as a result faster and stronger is the binding. The reduction in number of intra-molecular H-bond network for imidazole N atoms of Histidine in presence of monolayer is more pronounced (Appendix A) for H_3_ than H_1_, which can be a potential cause for making imidazole more available [54] for trans-esterification.

We observe that while both peptides prefer to remain in stretched conformation in absence of the cluster, the Au-MPC induces reduced flexibility, followed by a curved conformation, and a decrease in the number of intra-molecular hydrogen bonds especially for Histidine. All these changes are in line with an increase in the catalytic power of the bound peptide compared to the unbound one, in agreement with the experimental findings [41]. These changes become more prominent for the longer peptide with more Histidine; for the shorter peptide, formation of transient binding moieties by the functionalized ligands almost over the entire monolayer surface are more significant. On the other hand, for the longer peptide the binding is stronger and faster. Thus, we highlight a potential correlation between peptide length and increased rate constant of trans-esterification based on microscopic information using MD simulation that goes beyond the simple availability of a larger number of catalytic residues.

## 4. Materials and Methods

MD simulations at all-atom to coarse-grained scales are assessed tools for the study of non-static binding partners such as that involving a peptide (or protein) and a NP [55,56,57,58]. The difficulty in identifying the preferred orientation of biomolecules toward a given NP using MD simulations is due to the existence of many putative binding poses on the NP, nearly degenerate in energy, associated to the quite uniformity of the NP. We wish to remark that one of the main aims of the present study was that of disclosing the dynamics of the peptide association on the surface of Au-MPC. Hence, by performing metadynamics and/or other enhanced sampling simulations, we would have forced the dynamics of the system along a given collective variable, thus we would have lost a faithful temporal evolution of the binding association process. We proceed with 1 µs long MD simulation of peptides and nanocluster separately, followed by an unbiased sampling of different peptide conformations on top of Au-MPC. We sample the initial conditions by means of flexible Brownian Dynamics (BD) simulations, to remove biases that could lead to local minima in the binding process if simulated directly by atomistic MD. Additionally, a conformational sampling of the flexible peptide in water and during the association dynamics is taken into account by classical MD. This allows using MD simulations to predict properties of the peptide-Au-MPC complexes such as amino acid-specific interactions with NP ligands and supramolecular structure that ultimately drive the formation and the catalytic activity of peptide-NP conjugates.

Being aware that the results are affected by the quality of FF parameters used for the Au-MPC, we have developed ad hoc parameters for the system based on ab initio calculations and following the same strategy used to develop GolP [59] but taking explicitly into account the chemical nature of the ligand groups. Ligands are parameterized with OPLS/AA force field [60] and sulphur and gold atoms are parametrized as described before [61]. Gold polarization is neglected because the binding between the peptide and the Au-MPC is occurring through the ligand atoms of the monolayer and not directly through gold core atoms. The highly charged self-assembled monolayer makes the electrostatic interactions dominant, thus, polarization of gold atoms does not play a significant role in this kind of monolayer protected gold cluster.

The GolP FF has been originally developed for large bare gold NP (modelled ad flat surfaces) [59] and firstly extended to citrated capped gold NP [62]. The parameterization has been proved to be able to reproduce NMR chemical shifts of large proteins, e.g., Ubiquitin and β2-microglobulin, on citrate-capped gold NP [62,63,64]. The FF has been further tested and validated on the absorption of β2-microglobulin on differently functionalized small gold NPs [62,63].

In this work, a multi-step computational pipeline is used that includes: (i) 1 μs MD simulations of the peptide in water (without the Au-MPC) providing an initial extensive conformational sampling, crucial for the description of the relevant phases of the peptide adsorption process, (ii) flexible Brownian Dynamics (BD) including multiple flexible conformations of the peptide and a rigid body representation of the Au-MPC in implicit solvents, (iii) 1.5 μs MD refinement simulations of the most relevant peptide-Au-MPC complex resulting from the clustering analysis of the BD trajectories. The protocol is based on different software packages implemented in a compatible way at the different scales, i.e., the SDA7.2.2 software [42] and the GROMACS package [65] for MD simulations using the newly developed Au-MPC FF within extended GolP [59].

### 4.1. Set-Up of the H_1_ and H_3_ Peptide Systems

Two peptide sequences, namely, HWDDD (H_1_) and HHHWDDD (H_3_), are simulated in aqueous solvent. The initial structure of the peptides is obtained using Swiss-PDB Viewer which is the graphical counter part of the Swiss Model repository [66]. The 3D model of the target sequence is obtained after aligning it over a known structure of sequence similarity, followed by energy minimization to obtain the secondary structural assignment. The initial structure of the peptide H_1_ is shown in Appendix A. OPLS/AA force field parameters [60] are used for the peptide in its neutral protonation state, SPC/E water model [67] is used within GROMACS-5.0.4 version [65,68] in a cubic box with a 4 nm side, using periodic boundary condition [45,46]. The system contains 5186 atoms; 3 Na^+^ ions are added to guarantee the neutrality of the system. Initially, the entire system is minimized for 50,000 steps using steepest descent algorithm [69], followed by 100 picosecond (ps) isothermal isochoric (NVT) and 100 ps isothermal isobaric (NPT) equilibration. The Bussi-Donadio-Parrinello [70] thermostat is used to maintain temperature at 300 K and for pressure coupling at 1 bar Parrinello–Rahman [71] barostat is used. Simulation time step 2 femtosecond (fs) is chosen and Newton’s equation of motion is solved using leap-frog algorithm, short range cutoff for van der Waals and electrostatic interaction is ~1.0 nm, LINCS algorithm is used to constrain bond length and Maxwell Boltzmann distribution is considered to assign velocity [45,46]. Particle Mesh Ewald method [72] is used to take into account long ranged interaction, trajectory is saved every 10 ps. Finally, 1 µs simulation at neutral pH (we have used the standard protonation state of Histidine according to the experimental [41] pH = 7, as the pKa value of Histidine side chain is <7, we consider a neutral protonation state for the residue), room temperature and ~10 mM ionic strength is performed and the analyses are done over last 500 ns data. The same protocol is followed to build the structure of H_3_, namely HHHWDDD peptide (initial structure in Appendix A) and to perform 1 µs simulation trajectory.

### 4.2. Set-Up of the Nanocluster Model Au_144_(L)_60_ (L = S(CH_2_)_8_NH_2_^+^)

A 3D model for the functionalized gold nanocluster compatible with GROMACS engine is initially built with ‘NanoModeler’ [73] from which an initial geometry is obtained. The geometry is built from the webserver first assigning the correct morphology to core gold atoms Au144 and its staple-like motif (see Appendix A), which are connecting the S ligand atoms to gold surface (AUS) and gold interface (AUL) atoms [74]. Finally, the ligands are processed to form a homogeneously assembled monolayer. However, the resulting topology for the entire complex is generated using AMBER force field [74], where only non-bonded [75] interactions are incorporated for gold atoms without assigning charge or any bonding parameters. Thus, OPLS/AA FF parameters are developed for the Au-MPC compatible with the previously developed GolP FF [61]. For the entire Au-MPC (Appendix A), OPLS/AA topology is generated using TPPmktop server [76] and LJ parameters for Au atoms are taken from GolP. Finally, RESP [77] (Restrained electrostatic potential) charges are explicitly derived for all the atoms belonging to the smallest Au-MPC surface repeating unit, namely AUS-RS−AUL−SR-AUS. Where SR is the alkanethiol functional groups connected to the core through a sulfur atom forming the “staples” (see Appendix A), S is the sulfur atom forming a covalent bond with one gold atom at the interface (AUL = gold ligand) and a gold atom at the surface (AUS = gold surface). The charges are derived using R.E.D. server [78]. FF parameters are derived according to a strategy already used for different gold NPs which has been already extensively tested [61,79].

A cubic simulation box of dimension 6 nm is built and neutralized by adding 60 CL. Gold core atoms are kept frozen during equilibration and production run to maintain integrity of the structure, while the interfacial surface atoms are relaxed. Nose-hoover Thermostat [80] is used to achieve NVT equilibration, namely, 1 ns NVT equilibration is performed by keeping the entire nanocluster fixed (to capture electrostatic screening generated by ions [81]), to allow the ions to relax around the NP monolayers in water-box and then the self-assembled layer is freely relaxed for another 1 ns including ions and water molecules for temperature coupling. One µs production run is performed and post-processing analyses are performed on the trajectories collected in the last 500 ns.

### 4.3. Set-Up of the Flexible Docking Simulations

The flexible docking used in this work and implemented on SDA7.2.2 software [42] allows more than one rigid structure of the peptide to be included, thus incorporating peptide conformational flexibility into the BD simulations. The initial structures of peptides H_1_ and H_3_ to be included in the flexible docking are extracted from the 1 µs MD trajectories after clustering analysis (details in Section 2.1). Thus, three different conformations of H_1_, two for H_3_ and one for nanocluster (rigid conformation) are chosen to start the Brownian Simulation including flexible docking within SDA7.2.2. code [42,82].

Total interaction energy ($U_{Repr})\;$ between two solutes is computed as the summation of (a) long-range electrostatic interactions ($U_{EP})\;$ between the pair, (b) component associated with short range electrostatic desolvation ($U_{ds}^{e}$), (c) non-polar desolvation resembling hydrophobic collapse ($U_{ds}^{h}$) (More details in Appendix A).

A grid dimension of 161 × 161 × 161 ${Å}^{3}$ is used to build the electrostatic potential grid with a grid spacing of 1.0 Å using APBS program [83], including 10 mM ionic strength, the dielectric constant of the water solvent (78.0 at room temperature ~300 K) and of protein interior (2.0). For each system, 5000 runs are performed to identify peptide preferred binding orientation on the Au-MPC and the corresponding trajectories are collected, each with 500 ns simulation time. Typically, this type of simulation is performed within a box of spherical symmetry, at first the center of one solute is placed at 100 Å distance from the center of the other solute. Simulation time step is chosen 1 ps as long as the center-to-center distance between those two solutes remains <50 Å. For the nanocluster terminal N atoms and for the peptide O atoms of side chain of residue D, they are considered to calculate effective charges. The translational diffusion coefficient for the peptide is set to 0.27 $\times \;10^{-5}$
${\mathrm{cm}}^{2}$/s and for the gold nanocluster is set to 0.11$\;\times \;10^{-5}$
${\mathrm{cm}}^{2}$/s according to HYDROPRO [84] software, whereas rotational diffusional coefficients are 30.7 $\times \;10^{-5}{\;\mathrm{rad}}^{2}/\mathrm{ps}$ and 2.1 $\times \;10^{-5}{\;\mathrm{rad}}^{2}/\mathrm{ps}$ for the peptide and the cluster, respectively. The 1000 lowest energy configurations are saved, two different docking positions of the solute are considered based on a threshold root mean square deviation (RMSD) 1 Å between these two configurations. For H_3_, the translational diffusion coefficient is set to 0.25 $\times \;10^{-5}$
${\mathrm{cm}}^{2}$/s and the rotational diffusional coefficient 22.8 $\times \;10^{-5}{\;\mathrm{rad}}^{2}/\mathrm{ps}$.

### 4.4. Set-Up of the MD Refinement of Supramolecular Complexes

The most populated and energetically favourable peptide-Au-MPC association complexes resulting from the flexible docking are refined by 1.5 µs MD. At first, the peptide center of mass is translated by 5 Å with respect to the center of mass of the Au-MPC, without altering the binding orientation of the docking pose. Following our previous studies [61], the procedure is employed to avoid kinetic trapping of the peptide on the surface of Au-MPC, which can lead to the lack of a proper relaxation of the peptide over the surface. For peptide H_1_, a cubic simulation box is used with dimension set to 7 nm, and 34,382 atoms including 57 Cl^−^ ions are added. All the other technical details of the simulation are the same as those used for simulations of Au_144_(L)_60_ cluster in aqueous solvent. A gradual NVT equilibration is performed, at first only ions and water molecules are relaxed, then also interfacial ligands atoms are relaxed and finally the entire system is freely relaxed in explicit solvent (keeping core gold atoms frozen) for 1 ns. Finally, 1.5 µs simulation of production run are performed. The same MD refinement protocol is applied to the H_3_-Au-MPC complexes resulting from docking. A cubic box of dimension 7 nm was used with a total number of atoms equal to 37,513. Next, 1.5 µs simulation is performed and followed by the analysis.

### 4.5. Analysis of MD Trajectories

We used Visual Molecular Dynamics (VMD) [50] and PyMoL [85] software to visualize trajectories over simulation time. RMSD is computed to ensure equilibration of the simulation. SASA, radius of gyration, pair correlation function, mean square displacement (MSD) and cluster analysis are some of the parameters that are being analysed (details in Appendix A). All other calculations to extract information about microscopic properties of the systems are computed using in house codes.

## 5. Conclusions

In the present study, we have shown how a computational protocol that involves development of new atomistic force field parameters, flexible docking with Brownian Dynamics and µs-long MD simulations can be useful to gain the microscopic picture at the basis of experimental results for complicated bio-nanosystems such as nanozymes. In particular, in our study, we emphasized the collective effects of peptide sequence length, peptide conformation, dihedral flexibility, underlying structure dynamics of binding mode between peptide-Au-MPCs, and the potential role of H-bond network in modulating catalytic activity of this supramolecular assembly. The computational methodology described here can be useful for other problems within this class, such as to tune not only catalytically active peptide-nano conjugate but also to design proper nanoreceptor with target affinity in domain of nanomedicine and industrial applications in future.

## Figures and Tables

**Figure 1 ijms-22-03624-f001:**
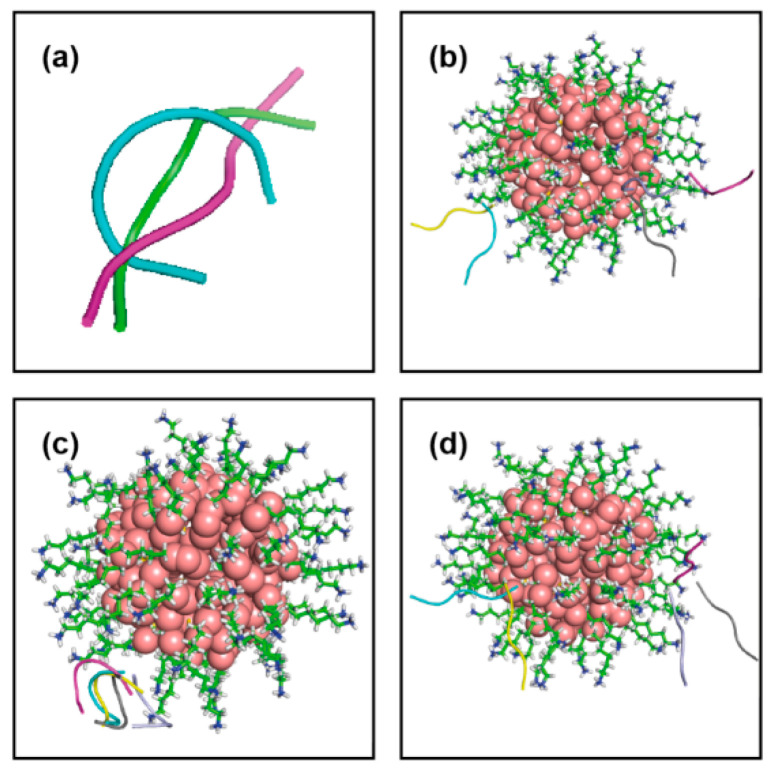
(**a**) Three different conformations of shorter peptide, H_1_. (**b**) Different docking poses obtained for first conformation of H_1_ with Au_144_(L)_60_, gold: sphere, functionalized ligand: stick, cyan:cluster1, magenta:cluster2, yellow:cluster3, light blue: cluster4 and gray:cluster5 and same for (**c**) second conformation, followed by (**d**) third conformation.

**Figure 2 ijms-22-03624-f002:**
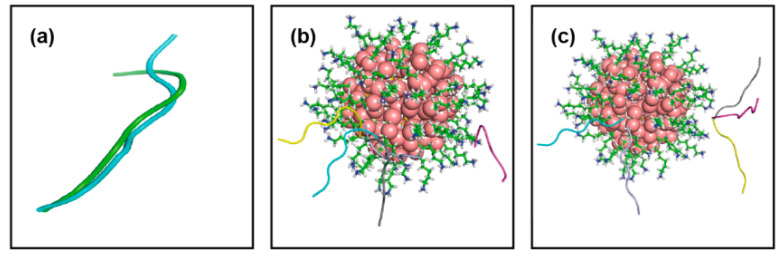
(**a**) Two different conformations for longer peptide, H_3_ generated using cluster analysis over simulation. (**b**) Different docking poses obtained for first conformation of H_3_ with Au_144_(L)_60_ and that for (**c**) second conformation, color code is the same as in Figure 1.

**Figure 3 ijms-22-03624-f003:**
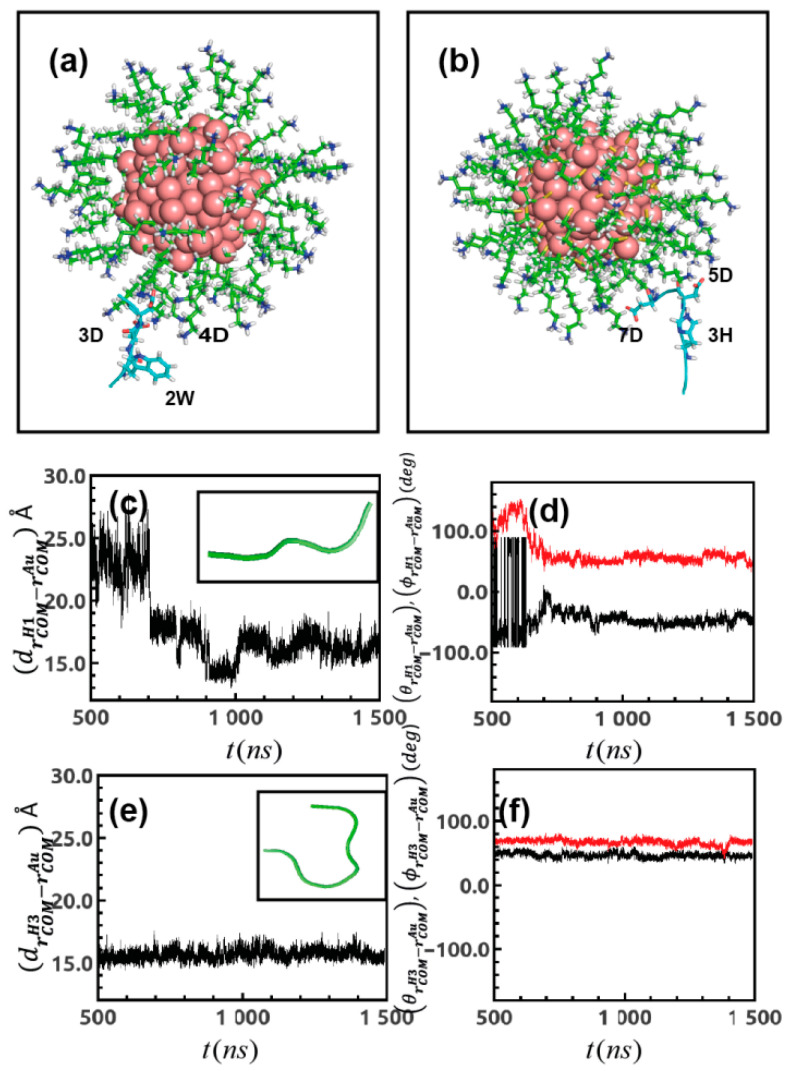
(**a**) Docked structure of H_1_ with Au_144_(L)_60_ and that for (**b**) H_3_ with Au_144_(L)_60_ as obtained from SDA and chosen for MD refinement; contact residues are shown in stick representation (cyan). (**c**) Distance fluctuation between center of mass of H_1_ and gold-cluster $\left(d_{r_{COM}^{H1}-r_{COM}^{Au}}\left(t\right)\right)\;$ over simulation time; inset: most popular conformation of the peptide in presence of gold-cluster as generated from equilibrated trajectory. (**d**) Angular representation of $\left(d_{r_{COM}^{H1}-r_{COM}^{Au}}\left(t\right)\right)$ as function of *t*. (**e**) $\left(d_{r_{COM}^{H3}-r_{COM}^{Au}}\left(t\right)\right)$ vs *t* plot; inset: most crowded conformation of longer peptide H_3_ generated after simulation performed in presence of gold-cluster and (**f**) angular component of $\left(d_{r_{COM}^{H3}-r_{COM}^{Au}}\left(t\right)\right)\;$ over simulation time.

**Figure 4 ijms-22-03624-f004:**
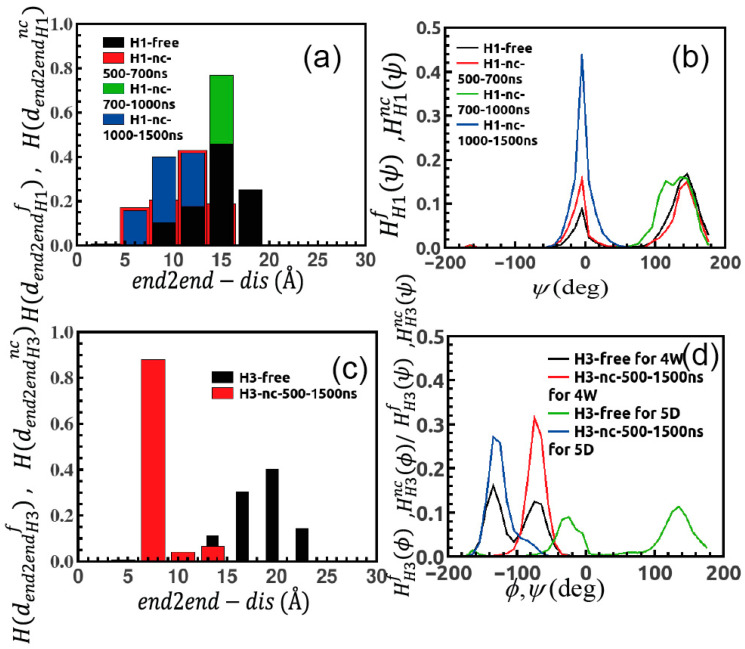
(**a**) Distribution of variation of end-to-end distance of H_1_ in free and in presence of gold-cluster; peptide is less stretched in bound form and that for (**b**) distribution of dihedral fluctuation, peptide loses its flexibility in presence of gold cluster. (**c**) Histogram of end-to-end distance for H_3_ showing peptide achieves curved conformation in complex state. (**d**) Similarly, dihedrals become less flexible in proximity with nanocluster.

**Figure 5 ijms-22-03624-f005:**
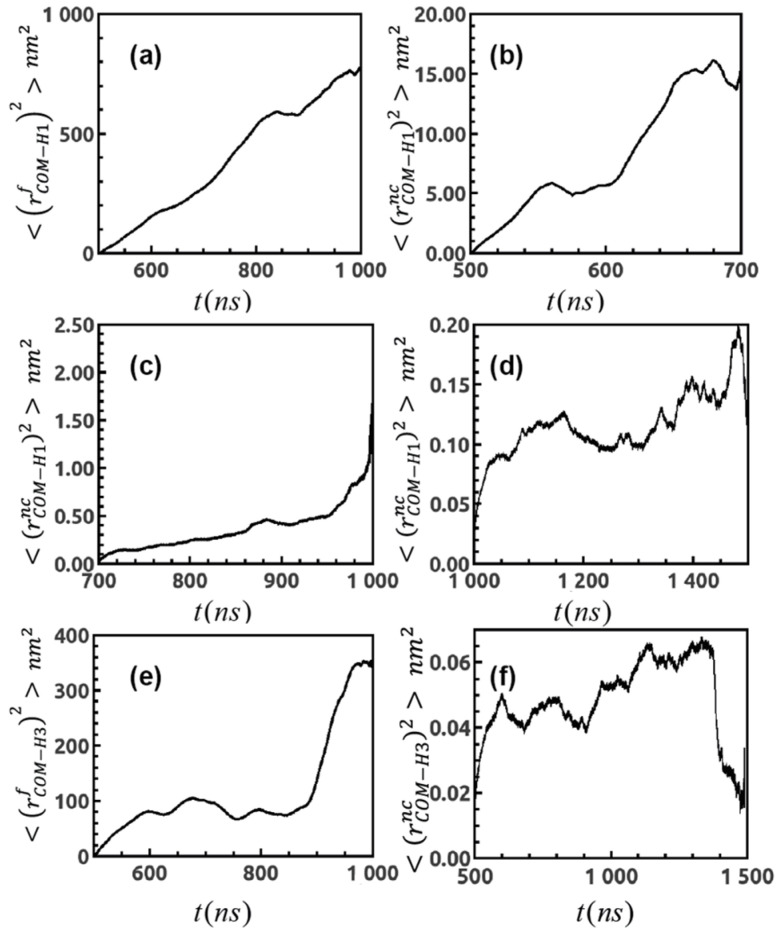
(**a**) MSD plot for center of mass for peptide H_1_ in free state, indicating normal diffusive behavior and that for in presence of nanocluster in (**b**) for 500–700ns, (**c**) 700–1000 ns, (**d**) 1000–1500 ns demonstrating deviation from linear dependence of MSD over time. (**e**) MSD plot for center of mass for peptide H_3_ in free and (**f**) in presence of gold cluster signifying signature of binding.

**Figure 6 ijms-22-03624-f006:**
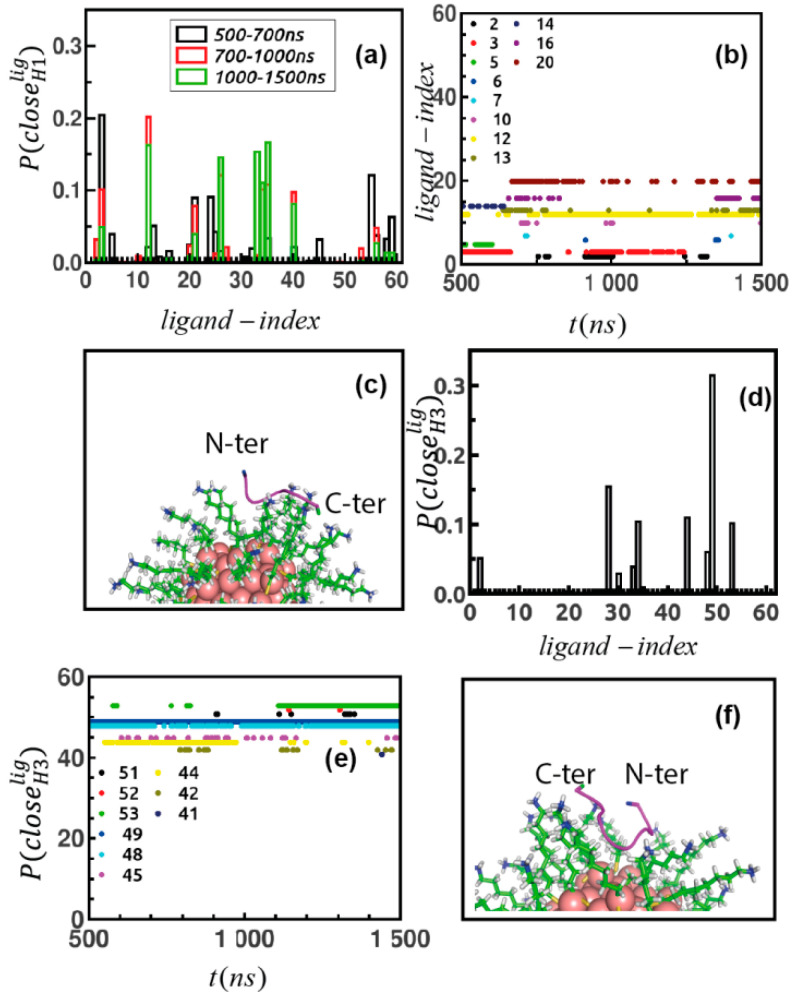
(**a**) Statistical plot of ligands showing strong contribution in binding with peptide H_1_. (**b**) Ligands that make contact with peptide H_1_ simultaneously over simulation time. (**c**) Final binding orientation of H_1_ bound Au_144_(L)_60_; ligand in cyan, stick mode, peptide in magenta, binding residue terminal ASP are in red and N Histidine is in blue, CPK representation. (**d**) Contributing ligands in binding with H_3_. (**e**) Ligands making simultaneous contact with H_3_ over simulation time and (**f**) final structure of H_3_ with Au_144_(L)_60_, color code is same as Figure 6c.

**Table 1 ijms-22-03624-t001:** Peptide (H_1_)-nanocluster docking results as obtained from SDA.

Conf Index	Clust Index	RelPop (%) ^(a)^	$\left(U_{Repr}\right)$ kT ^(b)^	$\left(U_{EP}\right)$ kT ^(c)^	$\left(U_{ds}^{e}\right)$ kT ^(d)^	$\left(U_{ds}^{h}\right)$ kT ^(e)^	Spread ^(f)^
**1**	**A1**	**70.8**	− **52.815**	− **50.607**	**8.886**	− **11.094**	**1.203**
1	A2	12.0	−51.375	−51.288	9.046	−9.132	1.172
1	A3	7.0	−50.788	−50.638	9.936	−10.085	0.571
1	A4	8.4	−51.005	−51.168	9.391	−9.227	0.343
1	A5	1.8	−51.006	−49.864	9.771	−10.913	0.495
**1**	**B1**	**63.5**	− **50.890**	− **47.567**	**8.618**	− **11.841**	**0.342**
1	B2	18.9	−50.631	−45.628	7.283	−12.287	0.500
1	B3	11.4	−50.676	−45.877	6.823	−11.622	0.464
1	B4	3.8	−50.268	−47.143	8.347	−11.472	0.254
1	B5	2.4	−50.361	−48.345	8.667	−10.683	0.273
**1**	**C1**	**37.3**	− **51.075**	− **50.940**	**10.970**	− **11.105**	**0.626**
1	C2	21.7	−51.325	−49.756	9.965	−11.534	0.518
1	C3	33.9	−50.266	−49.160	10.833	−11.940	0.601
1	C4	5.6	−50.346	−50.611	10.391	−10.126	0.387
1	C5	1.5	−50.230	−50.292	10.715	−9.654	0.001

(a) Relative population of this cluster (b) $U_{Repr}$: total interaction energy of the representative of the given cluster in kT with T = 300 K, (c) $U_{EP}$: total electrostatic energy of the representative complex, in kT, (d) $U_{ds}^{e}$: electrostatic desolvation energy of the representative complex, in kT, (e) $U_{ds}^{h}$: hydrophobic desolvation energy of the representative complex, in kT, (f) Spread: RMSD of the structures within the cluster with respect to the representative complex, Å.

**Table 2 ijms-22-03624-t002:** Peptide (H_3_)-nanocluster docking results as obtained from SDA.

Conf Index	Clust Index	RelPop (%) ^(a)^	$\left(U_{Repr}\right)$ kT ^(b)^	$\left(U_{EP}\right)$ kT ^(c)^	$\left(U_{ds}^{e}\right)$ kT ^(d)^	$\left(U_{ds}^{h}\right)$ kT ^(e)^	Spread ^(f)^
**1**	**D1**	**78.5**	− **55.104**	− **51.538**	**9.048**	− **12.613**	**1.271**
1	D2	10.7	−53.506	−52.317	10.716	−11.905	0.821
1	D3	3.6	−53.496	−52.045	9.897	−11.347	0.369
1	D4	3.7	−53.642	−50.904	10.819	−13.557	0.292
1	D5	3.5	−53.541	−49.327	9.270	−13.484	0.267
**1**	**E1**	**44.4**	− **54.087**	− **52.973**	**10.398**	− **11.511**	**2.127**
1	E2	26.2	−54.218	−53.461	9.214	−9.971	1.573
1	E3	15.8	−53.681	−54.362	10.045	−9.364	4.559
1	E4	10.8	−54.442	−51.790	8.913	−11.565	0.578
1	E5	2.8	−53.662	−53.151	9.899	−10.410	1.772

(a) Relative population of this cluster (b) $U_{Repr}$: total interaction energy of the representative of the given cluster in kT with T = 300 K, (c) $U_{EP}$: total electrostatic energy of the representative complex, in kT, (d) $U_{ds}^{e}$: electrostatic desolvation energy of the representative complex, in kT, (e) $U_{ds}^{h}$: hydrophobic desolvation energy of the representative complex, in kT, (f) Spread: RMSD of the structures within the cluster with respect to the representative complex, Å.

## Data Availability

The study did not report any data.

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
