# Peer review of "Molecular Dynamics Simulations of a Catalytic Multivalent Peptide–Nanoparticle Complex"

_ijms, 2021, doi:10.3390/ijms22073624_

Round 1

Reviewer 1 Report

The work by Dutta et al. is a very systematic study of the peptide-nanoparticle complex that is describe by molecular dynamics (MD) simulation, molecular docking and Brownian dynamics (BD). The study show an interesting correlation in the space of amino acids length (i.e. H1 and H3 with 5 and 7 aa) and a reduced flexibility during binding the nanoparticle that may highlight some degree of stability in the complex. I find the study theoretically consistent but I am afraid the article could reach a broad audicen if the author try to discuss some experimental evidence regarding their findings as it was mentioned in the abstract (i.e. for techno-logical and nano-medical applications). Before fully supporting for publication I will ask to address major comment below:

Major comments:

I can not find any discussion about short comings of the FF. I might have been parametrized well-enough for AU/Peptide complex, but what about very lengthy (above 7 aa) peptide/protein. I may suggest to take a look into a more versatile FF readily available (e.g. Gō-MARTINI https://doi.org/10.3389/fmolb.2021.619381) for modelling of large conformational changes in protein complexes and discuss/place in perspectives.  

Regarding the binding and the reduce flexibility, I would expect author to discuss the degree of stability of their peptide conformations. One can calculate the secondary structure composition for each chain once it binds the NP. This study can highligh whether a peptide follows a coil-like behaviour or it gets structured. In addition, one can briefly compute the number of native contact established by the peptide that stabilise a given conformation. In this way, one can extract mechanical stability as in ref. for protein (https://doi.org/10.3762/bjnano.10.51, server are available for contact map determination: http://pomalab.ippt.pan.pl/GoContactMap/) out of the final poses. This will certainly justify the reduce flexibility observed in some peptides.

 Minor comments:
Line 2-1: Please either use case sensitive for all words
Line 12: typo or define s-long 
Line 50: remove capital in Sulphur
Line 59: write molecular dynamics simulation (MD)---> molecular dynamics (MD) simulation, here and in other instances
Line 149: missing full stop
Line 178: Why authors do not consider water dielectric contact as 80 at room temperature 23 C?
Line 210: typo 37513. 1.5 

Author Response

Reply to Reviewer 1

The work by Dutta et al. is a very systematic study of the peptide-nanoparticle complex that is describe by molecular dynamics (MD) simulation, molecular docking and Brownian dynamics (BD). The study show an interesting correlation in the space of amino acids length (i.e. H1 and H3 with 5 and 7 aa) and a reduced flexibility during binding the nanoparticle that may highlight some degree of stability in the complex. I find the study theoretically consistent but I am afraid the article could reach a broad audience if the author try to discuss some experimental evidence regarding their findings as it was mentioned in the abstract (i.e. for techno-logical and nano-medical applications). Before fully supporting for publication I will ask to address major comment below:

We thank the Reviewer 1 for the positive appraisal of our work. We have followed all his/her suggestions and we have implemented all the recommendations through the text.

As some experimental evidences supporting our theoretical findings, we wish to remark that the present study represents a “proof of concept” of the design of nanozymes for catalytic applications which has been already studied experimentally. In fact, we have mentioned in the manuscript, the experimental study by Zaramella, D., Scrimin, P., and Prins, L.J.: ‘Self-assembly of a catalytic multivalent peptide-nanoparticle complex’, J Am Chem Soc, 2012, 134, (20), pp. 8396-8399 namely, ref. [41] along the manuscript.

For the sake of clarity, we have added the following sentence on Page. 2 of the Introduction:

“Here we investigate this system using atomistic simulations.”

After the sentence:

“Experimentally, a linear correlation was noticed between the increase in the catalytic rate constant and the number of Histidine present in the sequence (H1-3)[41] suggesting the importance of peptide sequence length in modulating catalytic activity of this kind of multivalent system.”

Major comments:

I cannot find any discussion about short comings of the FF. It might have been parametrized well-enough for AU/Peptide complex, but what about very lengthy (above 7 aa) peptide/protein. I may suggest to take a look into a more versatile FF readily available (e.g. Gō-MARTINI https://doi.org/10.3389/fmolb.2021.619381) for modelling of large conformational changes in protein complexes and discuss/place in perspectives.

We are confident that the large number of references ref. [42], [45], [46], [48], [49] and [50] already cited along the manuscript can satisfy the reviewer’s concerns about the validity of the atomistic FF developed and used in this work. For the sake of clarity, we have added the following sentence in paragraph 3 of section 2:

“The GolP FF has been originally developed for large bare gold NP (modelled as flat surfaces) [46] and firstly extended to citrated capped gold NP[49]. The parameterization has been proved to be able to reproduce NMR chemical shifts of large proteins, e.g. Ubiquitin and β2-microglobulin, on citrate-capped gold NP[49-51]. The FF has been further tested and validated on the absorption of β2-microglobulin on differently functionalized small gold NPs[49, 50]. “

We thank the reviewer for the suggestion of considering Gō-MARTINI FF, the corresponding paper is now cited as new ref. 84. However, its parametrization is based on a combination of top-down and bottom-up techniques and not on ab-initio calculations. Gō-MARTINI FF represents an assessed lower resolution model to study proteins but it has only recently extended to other systems such as Au-MPC. We are currently developing ad hoc Coarse Grained parameters for the Au-MPC based on the fully atomistic MD simulations discussed in this manuscript, in line with our FF development philosophy, which will be the subject of a future publication.

Regarding the binding and the reduce flexibility, I would expect author to discuss the degree of stability of their peptide conformations. One can calculate the secondary structure composition for each chain once it binds the NP. This study can highlight whether a peptide follows a coil-like behaviour or it gets structured. In addition, one can briefly compute the number of native contact established by the peptide that stabilise a given conformation. In this way, one can extract mechanical stability as in ref. for protein (https://doi.org/10.3762/bjnano.10.51, server are available for contact map determination: http://pomalab.ippt.pan.pl/GoContactMap/) out of the final poses. This will certainly justify the reduce flexibility observed in some peptides.

We are grateful to the reviewer for his/her suggestions. We have followed the suggestion and we have applied the secondary structure analysis and the contact map analysis highlighting interesting propensities of the two peptides.

We have added the results in Fig. S9 and S10 of the Supporting Information, we are citing the reference mentioned here by the reviewer as new ref. [82-83] and we have added the following discussion on Pag. 15:

“The secondary structure content for the free peptide in solution and for the peptide complexed with the nano-cluster has been computed using the DSSP built-in algorithm of (VMD)[75]. Figures S9(a)-(d) indicate that the overall effect of the Au-MPC on both peptides is to increase the β-turn propensity with respect to the random coil. This effect is more prominent in the case of H3. We then compute the stability[82] of the peptides in terms of the number of contacts in the final conformations and in different environments using Go̅ contact map analysis[83, 84]. The Go̅ coarse grained model is based on a lower resolution description of the peptide in which beads are located at Cα position. For the present application, a fully atomistic force field represents a very good balance between accuracy and computational feasibility. For longer peptides, simplified schemes such as coarse grained descriptions (e.g., the recent Go-MARTINI FF for peptide) should also be considered, although such FFs should be extended to the nano-cluster as well. The analysis is based on contact generated from native structure and interacting through LJ potential and it represents an established method to understand conformational transition in proteins. When the method is applied to our peptides, we find the presence of close contacts only between Cα atoms of residues 1H and 6D of H3 peptide in the presence of nano-cluster, based on the standard distance threshold and sequence distance cut-off. No contact is observed in the corresponding free state of the same peptide. This implies the presence of significant changes in the peptide conformation occurring only in the longer peptide sequence and its bound form.”

Minor comments:

Line 2-1: Please either use case sensitive for all words

We agree with the reviewer and we have made the changes in Title appropriately.

Line 12: typo or define s-long 

We have modified the typo ‘s-long’ by ‘µs-long’ in the Abstract.

Line 50: remove capital in Sulphur

We fixed the capital letter accordingly.

Line 59: write molecular dynamics simulation (MD)---> molecular dynamics (MD) simulation, here and in other instances.

We have incorporated the change (see first line of third paragraph of Introduction section).

Line 149: missing full stop

We modify this properly (see Line number 18 in section 2.2).

Line 178: Why authors do not consider water dielectric contact as 80 at room temperature 23 C?

We decided to use the default dielectric constant of 78 for the water solvent and the room temperature 300 K, according to experiments.

Line 210: typo 37513. 1.5

We take care of this typo (see last line of section 2.4).

Reviewer 2 Report

In this paper, authors described the interaction of short peptides with monolater-capped gold nanoparticles using a computational approach. Their results indicate that short peptides are able to bind over the layer of ligands covering the nanoparticle, but the specificity of the binding depends on the peptide length and composition. The study is well performed and conclusions are supported by simulation data. I have some minor concerns:

  • in the methods section there are some unclear points: the GolP FF has been designed as surfaces and there are different kind of atoms representing the polarizability. In this case, the nanoparticle is spherical and it is not clear how authors applied the GolP FF to this particular system.
  • Lines 96-98: do authors used only classical MD without any enhanced sampling technique such as metadynamics to sample the peptide configurations? Why?
  • lined 156-157 and 205-206: for the peptide authors used the NPT ensemble, while not for the nanoparticle. Why this choice? there are problems in the pressure when adding the nanoparticle?
  • lines 81-82, refs [28-31], probably too many self-citations. Please insert citations about other groups instead (R. C. Van Lehn's work on the monolayer capped gold nanoparticles, A. Kyrychenko's works on silver nanoparticles, Petr Kral's work on absorption of peptides over gold nanoparticles, F. Tavanti's work on the absorption of proteins, H. Lopez for coarse grained model of nanoparticle-proteins interactions). These authors contributions (in particular those of Van Lehn and Kyrychenko) should be cited in the introduction to give to the reader a more clear view of previous works for example in lines 49-54.
  • The protonation state of His residues. In line 114, authors stated that they used a neutral protonation state. This point is trivial. Why neutral? 
  • Did authors performed a DSSP calculation on the secondary structure of the peptide?
  • Figure 6 c) and f) are not clear because they are too small and the orientation of the peptide is not clear. Please add some labels such as the N-terminal or similar to identify the amino acids.

Round 2

Reviewer 1 Report

I do appreciate the effort carried out by author during implementation of my comments. Thus, I endorse the study for publication and wish successes in future works.